# Effects of Dominance and Sprint Interval Exercise on Testosterone and Cortisol Levels in Strength-, Endurance-, and Non-Training Men

**DOI:** 10.3390/biology11070961

**Published:** 2022-06-24

**Authors:** Grzegorz Zurek, Natalia Danek, Alina Żurek, Judyta Nowak-Kornicka, Agnieszka Żelaźniewicz, Sylwester Orzechowski, Tadeusz Stefaniak, Magdalena Nawrat, Marta Kowal

**Affiliations:** 1Department of Biostructure, Faculty of Physical Education and Sport, Wroclaw University of Health and Sport Sciences, 51-612 Wroclaw, Poland; grzegorz.zurek@awf.wroc.pl; 2Department of Physiology and Biochemistry, Faculty of Physical Education and Sport, Wroclaw University of Health and Sport Sciences, 51-612 Wroclaw, Poland; 3Institute of Psychology, University of Wroclaw, 50-529 Wroclaw, Poland; alina.zurek@uwr.edu.pl (A.Ż.); sylwester.orzechowski@uwr.edu.pl (S.O.); magdalena.nawrat@uwr.edu.pl (M.N.); marta7kowal@gmail.com (M.K.); 4Department of Human Biology, Faculty of Biological Sciences, University of Wroclaw, 50-137 Wroclaw, Poland; judyta.nowak@uwr.edu.pl (J.N.-K.); agnieszka.zelazniewicz@uwr.edu.pl (A.Ż.); 5Department of Sport Didactics, Faculty of Physical Education and Sport, Wroclaw University of Health and Sport Sciences, 51-612 Wroclaw, Poland; tadeusz.stefaniak@awf.wroc.pl

**Keywords:** acute exercise, hormonal response, saliva

## Abstract

**Simple Summary:**

Exercise is a powerful stimulus to the endocrine system, modifying plasma concentrations of many hormones, including testosterone and cortisol, which are often used to describe fatigue in sport. In our investigation, we wanted to explore the hormonal response (testosterone and cortisol) in saliva after acute exercises in men who perform endurance-training and strength-training exercises compared to a non-training group. Participants performed sprint interval exercise. During the whole exercise, the participants’ heart rates were measured, and a rating of perceived exertion was assessed immediately after each bout. The study showed that there were no differences in testosterone and cortisol changes in the endurance-training, strength-training, and non-training groups after the sprint interval exercise. We suppose that one session of the sprint interval training should have more volume (more or longer duration of sprints) to provoke testosterone and cortisol reaction in endurance-training and strength-training individuals. However, the heart rates after acute exercise in the endurance-training and strength-training groups were lower than in participants from the non-training group.

**Abstract:**

The aim of the study was to investigate the response of testosterone and cortisol to sprint interval exercises (SIEs) and to determine the role of dominance. The experiment was conducted in a group of 96 men, divided into endurance-training, strength-training, and non-training groups. Participants performed SIEs consisting of 5 × 10-s all-out bouts with a 50-s active recovery. Using the passive drool method, testosterone and cortisol concentrations were measured in saliva samples at rest at 10 min pre and 12 min post exercise. Participants’ heart rate (HR) was measured during the whole exercise. Dominance was assessed by the participants before the study; the rating of perceived exertion (RPE) was measured immediately after each bout. The study showed that those who trained in endurance and strength sports had significantly lower mean HRs after five acute 10-s interval bouts than those in the non-training group (*p* = 0.006 and *p* = 0.041, respectively). Dominance has an inverse relation to changes in HR; however, it has no relation to hormone response. No significant differences were observed in testosterone and cortisol changes in the endurance-training, strength-training, and non-training groups after SIE (*p* > 0.05), which may indicate that the exercise volume was too low.

## 1. Introduction

Different exercise modalities and prior training experience (i.e., endurance training, resistance training, interval training), and variables within the modality (i.e., intensity, volume, duration), can result in different hormonal responses [1]. Knowing the hormonal responses after performing a single exercise session can indicate the direction of selecting an appropriate training strategy, which is important because of the great interest in endocrine responses due to the use of training modulation determining the level of changes in testosterone or cortisol concentrations [2]. Previous multifaceted analyses of current responses in a single interval exercise (SIE) session have focused primarily on determining changes in cardiorespiratory parameters that determine physical performance [3]. As reported by Riachy et al. [4], data on the effects of exercise on serum testosterone levels in men show considerable inter-individual and inter-study variability. This variability can be explained by (a) the use of different types of exercise (e.g., endurance, resistance, or interval training), (b) the intensity and duration of the exercise session, and (c) the fitness status of the participants. Testosterone is the main anabolic hormone, important for skeletal muscle growth and maintenance, as well as for neural function [5]. Cortisol, on the other hand, has catabolic effects [6]. Thus, the testosterone/cortisol (T/C) ratio is used as an indicator of the balance between anabolic and catabolic processes [7]. It is identified as an indicator of fatigue (decreased performance, psychological changes, and neuroendocrine disorders after some physical training) [8].

Additionally, changes in testosterone and cortisol concentrations interact to regulate dominance. In a study by Meht and Joseph [9], T was shown to be associated with dominance under conditions of threat or excessive challenge such as extreme physical exertion. Analyses by Batrinos [10] also showed that testosterone levels increase during the aggressive phases of sports games, which is determined by the level of dominance among athletes, while Carré and McCormick [11] indicated that there was no relationship between changes in T concentrations and dominance in the context of competition. Since dominance is associated with acquisition and a high sense of agency, these findings suggest that higher testosterone levels should only support a higher status when cortisol levels are low. When cortisol is high, higher testosterone levels may actually reduce dominance and, in turn, reduce the degree of competition or engagement in extremely hard efforts such as sprint interval exercise (SIE). However, this requires further analysis.

One of the most popular types of interval training is sprint interval training (SIT), which involves performing work at maximal intensity (generating the highest possible power, known as “all-out”) [2,12]. To understand the multiple mechanisms that regulate the state of physical adaptation, it is important to determine the hormonal responses to a single SIT session. The single session of SIT (sprint interval exercise—SIE) typically consists of two to six bouts lasting between 10 to 30 s, with recovery of a longer duration (e.g., up to several minutes), and a total duration of one session interval (SIE) of typically 10 to 30 min [13]. However, the 30-s maximal bouts used have been criticized due to negative affective reactions and reluctance to undertake subsequent interval sessions, despite their benefits in overall health improvement [14,15]. Thus, SIT with 10-s repetitions has recently received more interesting consideration given its effectiveness in cardiovascular and respiratory adaptations (increasing maximal VO_2_) and in skeletal muscle metabolism [16,17]. As was presented by Islam et al. [18], shorter sprints with more repetitions are perceived as more enjoyable and lead to a greater desire to get involved in SIT. Affect, intention, self-efficacy, pleasure, rating of perceived exertion (RPE), and preference were rated in favour of the shortest 5:40 protocol (24 × 5-s bouts, 40-s rest) versus (A) 30:240 (4 × 30-s bouts, 240-s rest); (B) 15:120 (8 × 15-s bouts, 120-s rest) [18]. However, they did not measure hormonal changes. According to Martinez-Diaz and Carrasco [19], among active male college students, a single interval session composed of 10 × 1-min sets of VO_2_ peak power output induced acute changes in mood states that seem to be associated with hypothalamic–pituitary–adrenal axis activation, as the magnitude of the cortisol response in the study reached 37% immediately after the interval session and as high as 77% 30 min after. In contrast, Beaven et al. [20], during maximal single repetition (1-RM) at intensities of 85, 70, 55, and 40% RM, showed no change in testosterone and cortisol.

Given the myriad of possible variants of SIE protocols and the small amount of research on their psychological perception, we chose the 10-s bout protocol and asked whether ratings of perceived exertion would differ depending on the type of activity being practiced (endurance, strength, or no activity in the control group) and questioned the role of domination. We also hypothesized that ratings of perceived exertion after SIE will be greater in the non-training group. In this way, factors capable of producing less-acute SIE protocols could be characterized. Therefore, the present study evaluated preference after an experimental 10:50 (5 × 10-s bouts, 50-s rest) SIE session protocol. The effect of SIE session on T and C levels during the rapid recovery phase in non-training, endurance-training, or strength-training individuals still remains unclear and requires further study. Therefore, the purpose of our study was to determine changes in T and C concentrations after a single SIE session in non-training, endurance-training, or strength-training participants. Analysis of C and T levels can provide valuable information about the physiological stress response and adaptation to one session of sprint interval training. Our initial hypothesis was that the level of T and C will be different depending on the characteristics of the exercise. In the group of non-training participants, the release of C will be higher; however, in the group of endurance- and strength-training participants, higher levels of T will be observed.

## 2. Materials and Methods

### 2.1. Participants

Ninety-six healthy men (aged 19–25 years; mean age ($\bar{x}$) = 21.25; SD = 1.79 years) participated in the study. Information about the project was posted on social media, and participants were also recruited through flyers and direct invitations from researchers during physical activity courses. The main criteria for inclusion in the study were generally good physical fitness and a lack of qualification to the risk group of cardiopulmonary and metabolic diseases; a lack of supplementation and/or hormone treatment; no injuries to the mouth and no dental orthodontic treatment during the experiment; and no consumption of alcohol and tobacco or other stimulants 24 h before the experiment. All participants were advised to brush their teeth at least 2 h prior to the study to minimize the impact on the hormonal assessment of saliva [21]. All participants were familiarized with the study procedure and gave written, informed consent to participate in the study. The ethical approval of the study protocol was provided by the Ethical Committee of the Institute of Psychology of the University of Wroclaw (date of the decision: 16 November 2020, decision number: 2020/ARDK), in accordance with the Helsinki Declaration.

None of the participants practiced sports at a professional level, while 30 participants declared the use of regular strength training (bodybuilding, CrossFit, resistant or “resistance” training), and 35 participants declared regular endurance training as a kind of basic training (running, swimming, soccer). Each participant exercised at least 3 times a week. Thirty-one non-training participants declared to perform 2–2.5 h of recreational irregular physical exercise (walking, cycling) twice a week—this made up the control group in our study. The groups were compared in terms of somatic parameters, that is, age (F(2.90) = 2.34, *p* = 0.103), body height (F(2.90) = 2.92, *p* = 0.059), and body mass (F(2.90) = 2.19, *p* = 0.117). Detailed characteristics of the participants are shown in Table 1.

### 2.2. Study Design

The testing session was conducted between 7.00 AM and 11.00 AM in order to avoid daily hormone fluctuations. Participants were instructed to maintain a sleeping pattern and dietary habits and to refrain from undertaking physical activity for 24 h before the testing session in the laboratory in order to reduce any bias in salivary T and C [21]. Before the testing session, participants filled out a questionnaire, reporting their age, how often they exercised, and the type of that activity, as well as the average duration of their trainings. Their body masses (kg) and heights (cm) were measured using a WPT 200 medical scale (RADWAG, Radom, Poland) before the physical exertion. BMI was calculated based on the participants’ weights and heights (body mass (kg) (height (m))^−2^). Heart rate (HR) was measured with the Polar S810 sport-tester (Polar Electro, Kempele, Finland) during all exercises. The HR measurement started two minutes before the warm-up and continued until one minute after the SIE. HR_mean_ is the average HR values from the start of the first repetition until the end of the fifth repetition, including recoveries. HR_rest_ is the averaged HR value of one minute of restitution after SIE.

The flowchart and study protocol is presented in Figure 1.

### 2.3. Salivary Hormone Analysis

Participants provided unstimulated saliva samples at rest 10 min before and 12 min after exercise [22]. T and C concentrations were measured in saliva samples, which were collected by participants using the passive drool method. Participants’ identification numbers and the words “pre” or “post” (exercise) were written on every collection tube. The samples were stored using standard procedures (−80 °C) until analysis [23]. Salivary measures of C and T were determined by competitive enzyme-linked immunosorbent assay (ELISA method). Before the analyses, samples were thawed and centrifuged for 10 min at 10,000 RPM. Clear supernatant was used for the quantitative determination of T and C by the commercial ELISA kit (DES6622 and DES6611, DEMEDITEC). The intra- and inter-assay variations for C were: intra: <6.8%; <9.4% with assay sensitivity 0.014 ng∙mL^−1^ for C, and for T <9.7%; <9.9% with assay sensitivity 2.2 pg∙mL^−1^. The calculation of the results was performed by constructing a standard curve (plotting the absorbance value of the standards (y-axis) against their concentration (x-axis)). Hormone concentrations in saliva samples were calculated in relation to a standard curve and expressed in ng for C and pg/mL for T.

### 2.4. Sprint Interval Exercise Sessions (SIE)

All participants performed one protocol of sprint interval exercise (Figure 2) on the cycle ergometer (Ergomedic Monark 894, Vansbro, Sweden), which followed the same scientific criteria as tools used in previous studies [24,25]. The physical exertion consisted of a 5 min warm up with a 2 kg load, 1 min of rest, 5 repeated “all-out” bouts of exercise (10 s each) with a Wingtate load—7.5% of the participant’s body mass (followed by 50 s of slow-cycling without a load between bouts), and 1 min of slow-cycling without a load at the end. Figure 2 displays the SIE protocol.

### 2.5. Scales Used

#### 2.5.1. Borg Ratings of Perceived Exertion (RPE) Scale

The RPE is a tool for the subjective assessment of exercise intensity, that is, the degree of fatigue in particular bouts. This scale allows respondents to relate the degree of fatigue during exercise to the fatigue experienced during everyday activities. In general, a score > 18 indicates that a maximal bout was made, and values > 15–16 indicate that the anaerobic threshold was exceeded. Ratings on this scale are related to HR. The principle of the scale is to divide the predicted heart rate for a given exertion by 10; hence, the exertion causing an increase in heart rate to 190 beats∙min^–1^ is scored 19 points, and the total rest in which the heart rate oscillates between 60–70 beats∙min^–1^ is scored 6–7 points. In the experiment, we used a shortened, 10-point version of the scale, where a score of 9–10 shows the almost maximal or maximal level of exertion [26].

#### 2.5.2. Dominance Scale

Dominance was measured with a 5-item questionnaire previously used in studies by Kowal et al. [27]. This scale has not yet been published. However, the aim of our previous investigations was to construct a reliable and validated scale to measure an individual’s dominance. The scale was found to be highly reliable (Cronbach’s alpha: 0.78), similar to the present study (Cronbach’s alpha: 0.753). The questionnaire included the following items: (1) ‘I often persuade others to behave as I suggest’; (2) ‘Everything usually turns out to be as I want’; (3) ‘I usually make decisions for myself and others’; (4) ‘It is rather me who influences others, and not the other way around’; (5) ‘I am dominant towards others’. Participants responded to each item on a 5-point Likert scale (range: 1—‘I definitely disagree’, 2—‘I disagree, 3—‘I don’t have an opinion’, 4—‘I agree’, and 5—‘I definitely agree’). In all subsequent analyses, we used a mean value of the dominance scale.

### 2.6. Statistical Analysis

In the first step, we computed participants’ BMI (body mass index (kg)/(height (m))^2^), HR_mean_ (a composite score of a mean of five measures of heart rate after five 10-s interval exercises), typical weekly physical activity (the number of trainings in a typical week × a typical length of training), and a mean score of self-reported dominance for each of the participants. Next, we subtracted pre-test testosterone and cortisol from post-test testosterone and cortisol to create testosterone and cortisol change indexes (respectively). In the next step, we calculated the Mahalanobis distance to screen for potential outliers (relying on the usually recommended criteria of *p* < 0.001) [28,29].

We then proceeded with the linear regression models. In the first model, we regressed HR_mean_ on the testosterone change, cortisol change, a mean score of dominance, and a type of physical activity performed by a given participant (i.e., endurance, strength, or other). In the subsequent models, we introduced (2) age, (3) typical weekly physical activity (number of trainings in a typical week × a typical length of training), and (4) BMI, and compared the models’ fit. We repeated the above steps, regressing both the cortisol and testosterone change on a type of physical activity performed by a given participant (i.e., endurance, strength, or other) and a mean score of dominance. All analyses were performed in Jamovi (1.8.1) and SPSS (Inc., Chicago, IL, USA).

## 3. Results

Detailed descriptive characteristics of the participants are shown in Table 1. Analysis of the Mahalanobis distance revealed four potential outliers, which we excluded from all subsequent analyses. When we compared the regression models, the first one showed a superior fit to the other models (*p* > 0.05). Thus, here, we report the results of the first model (see Table 2, Table 3 and Table 4). Results showed that individuals who trained endurance and strength sports had lower heart rate means following five acute 10-s interval exercises than individuals in the control group (while there were no differences between individuals who trained endurance and strength sports; see Figure 3). Moreover, dominance was negatively related, while the cortisol changes were positively associated with mean heart rates, meaning that those who reported being more dominant had a lower mean heart rate compared to those who reported being less dominant; also, the more acute the cortisol response, the higher the mean heart rate. Furthermore, the control group experienced a larger change in cortisol than the strength-training group, but there were no differences in the cortisol change between the endurance-training and strength-training and control groups (see Figure 4). Dominance was unrelated to the cortisol change, and the type of discipline and dominance were unrelated to the testosterone change.

## 4. Discussion

The purpose of this study was to determine the effect of a sprint interval exercise on changes in testosterone and cortisol levels and the T/C marker among non-training, endurance-training, and strength-training participants. The main results indicate that the SIE protocol performed by the endurance-training and strength-training participants does not cause significant differences in hormonal conditions compared to the control group, in which a linear relationship was demonstrated between the increase in cortisol concentration and the achievement of higher HR values. We also found that after SIE, there were statistically significant differences in changes of cortisol levels between the strength-training and non-training groups. Dominance has an inverse relation to changes in HR; however, it has no relation to hormone response.

Physical exercise is a particular form of activation of the hypothalamic–pituitary–adrenal (HPA) axis, providing an increase in cortisol levels [30]. The response of the HPA axis to exercise varies with the duration and intensity of exercise [31]. In contrast, the change in cortisol concentration is independent of individuals’ fitness status when exercise is performed at similar relative intensities among non-training and trained individuals [32]. Nevertheless, it is also accepted that endurance athletes have a reduced sensitivity to cortisol to protect muscle tissue during and after exercise. Endurance-training individuals show an adaptation of the HPA axis activity to repeated exercise due to reduced tissue sensitivity to glucocorticoids [32]. In their study, Luger et al. [30] indicated that highly trained runners had statistically significantly lower cortisol concentrations compared to sedentary people after prolonged exercise above 60% of maximal oxygen uptake (VO_2max_). In contrast, as reported by Dote-Montero et al. [4] in their meta-analysis, repeated-sprint training and sprint interval training, despite high intensity, may not be long enough to induce a strong increase in C levels in contrast to interval training bouts ≥ 60 s. This may explain the significant increases in the change of salivary C concentration in all groups after SIE (Table 1). This suggests that the particular exercise was not intense enough to elicit a hormonal response, making it advisable to measure, in future studies, the lactate concentration to determine, among other things, the intensity of the exercise. The value of such a study was shown in an article by Lu et al. [33], who noted that the surge in testosterone immediately following interval exercise is highly correlated with an increase in lactate concentration. Tanner et al. [34], in contrast to our study, showed a significant increase in C levels and obtained HR values close to maximal after a single interval session in trained individuals. However, as indicated by that study’s participants themselves, the exercise test was exhausting (six intervals of 3.5 min at a treadmill speed equivalent to 90% VO_2max_, interspersed by recovery periods of 2 min = at the speed equivalent to 30% VO_2max_), which could also affect the higher RPE values. In contrast to the results observed in that study, this difference with our findings can be explained by the exercise characteristics and the total duration of the protocol used: 5 × 10-s sprint interval exercise with 50-s recovery versus 6 × 3.5 min with 2-min recovery.

Endurance-training individuals have lower testosterone levels, which may be due to weight loss from this training [35] as opposed to strength-training peoples [36]. Additionally, Kreamer et al. [37] indicated that participants with 2 years of weightlifting training experience showed a significant exercise-induced increase in testosterone, while participants with training experience of less than or equal to 2 years showed no significant differences in testosterone change. Interestingly, Cadore et al. [38] observed lower reactivity of anabolic and catabolic hormonal responses in long-term strength-training men, indicating that higher training volume/intensity is required to induce significant hormonal changes. Statistically significant differences in C change observed between the non-training and strength-training groups can confirm the above-presented research (Figure 4).

The T/C is an appropriate indicator of an organism’s anabolic environment [8]. The role of testosterone in the body is to maintain anabolism through the process of protein synthesis. In contrast, cortisol has a catabolic function and is involved in the stress response. Most athletes aim to increase the T/C, thereby enhancing protein synthesis and tissue recovery after physical exercise [37]. However, in this study, we observed no significant differences in T/C after performing SIE in any of the study groups, suggesting that the exercise applied was not sufficient to increase the body’s anabolic environment. Similarly, individuals who trained both strength and endurance showed greater resistance to physiological stress by achieving a lower mean heart rate value after five repetitions of 10-s “all-out” exercise (Figure 3). Psychological reinforcement for this thesis is the fact that the dominance scale has an inverse relationship to changes in heart rate—the lower the HR, the higher the level of dominance (Table 1).

An explanation for the results obtained may be found in environmental psychology. Already, Seligman [39] has shown the importance of dominance as a feeling of control related to health and behavior. The adaptation to environmental demands expressed in the lower physiological parameters of our participants favors the increase of the psychological variable of dominance as expressing an internal control of one’s own behavior and health. According to Rivers and Josephs [40], dominance is as legitimate an environmental descriptor as pleasure and arousal.

Furthermore, results of Jiménez et al. [41] provided explanations for the lack of SIE effect in T, C, and T/C levels in our participants. After winning a league game, higher testosterone levels were observed in professional soccer players, compared to semi-professional or amateur athletes. In contrast, this temporary hormonal fluctuation was not observed after winning a friendly match or during a normal training day. In the same match, cortisol levels were lower in professional and semi-professional athletes compared to levels in amateur athletes. This means, in soccer players, the increase in testosterone was only noticeable when the team faced the real challenge of a league match. It follows that the desire to achieve a goal (and maintain social status) may be one of the key reasons why testosterone increases rapidly. Conversely, testosterone did not change after friendly games, suggesting that these situations are not true goals in which players do not perceive a real threat (in the sense of dominance) any more than they perceive the preparation for the next game in their daily training, or even in a friendly game. Thus, we speculate that the SIE we conducted did not present a real challenge to our participants in the sense of increased testosterone or dominance, nor did it trigger a stress response in the form of cortisol release. In our study, similar to Jiménez et al. [41], there was an adaptation to the exercise situation in the absence of both T and C output. Results of Jiménez et al. [41] show that cortisol levels were lower in professional and semi-professional athletes compared to those in amateur athletes. Again, similar to Jimenez et al. [41], we observed C changes after SIE between the non-training and strength-training groups. This allows us to suppose that the perception of SIE in groups of professional and amateur athletes depends not only on the type of activity performed, and that the changes in C concentration in amateur athletes are influenced by the intensity and volume of the exercise sessions undertaken.

Finally, some authors [14,15] have shown that SIE can elicit negative affective reactions during exercise, which is responsible for the withdrawal and avoidance of exercise in the future. Most importantly, however, the SIE exercise protocol appears to be perceived as too difficult and demanding in terms of the effort put into it, especially for individuals with sedentary lifestyles [15]. Our experiment did not involve assessing affect, intention, self-efficacy, enjoyment, and preference. However, we used the Borg scale to assess the intensity of rating of perceived exertion after SIE. We found no statistically significant differences in perceived exertion between the endurance, strength, and non-exercise groups (endurance 5.16 RPE; strength 5.60 RPE; control 5.36 RPE); thus, we did not confirm the hypothesis that the non-exercising lifestyle group had a significantly higher perceived exertion. Thus, following Islam et al. [18], a short 10-s sprint with 50-s of rest is perceived as enjoyable and leads to a greater desire to engage in SIE. We can speculate that SIE in our formulation may safeguard the physical activity of a healthy, non-exercising population of young people.

Our results should be interpreted with caution because lactate concentration was not measured in this study, which could enrich the interpretations regarding hormonal changes due to a single interval session. There is also no measurement of peak power output (PPO), work (W), or oxygen uptake (VO_2_), which could explain the mechanisms of C changes and the change in the physiological cost of the participants. For a better characterization of the participants, a more detailed analysis of the training experience should be conducted, especially regarding the volume and intensity of physical activity undertaken per week. The dominance scale has been validated, but not published to date, so we see this as a limitation in our study.

## 5. Conclusions

Sprint interval exercise consisting of 5 × 10-s “all-out” bouts performed by non-training individuals resulted in significant differences in cortisol changes concentrations compared to strength-training individuals. The lack of significant changes in T and C hormone concentrations among strength-training and endurance-training participants may indicate that the exercise volume was too low.

Non-training participants had higher HR after SIE than those from endurance- and strength-training groups; however, there were no differences in the ratings of perceived exertion. This suggests that it may be used by non-professional individuals.

## Figures and Tables

**Figure 1 biology-11-00961-f001:**
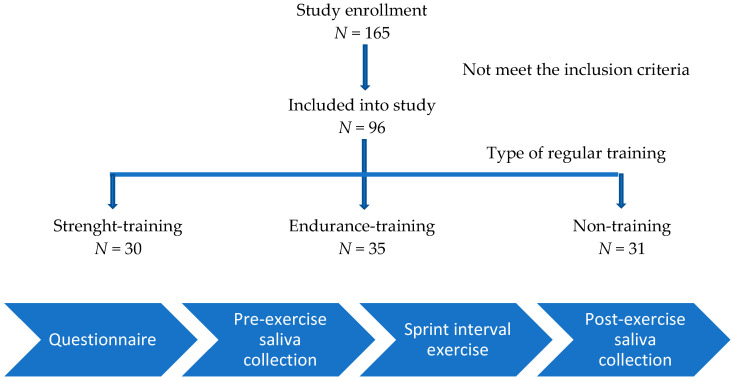
The flowchart and study protocol.

**Figure 2 biology-11-00961-f002:**
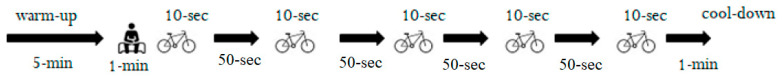
Sprint interval exercise protocol.

**Figure 3 biology-11-00961-f003:**
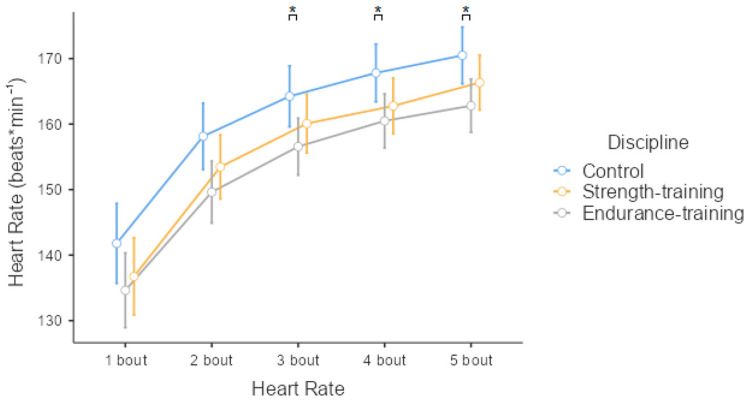
Means and confidence intervals (95%) of the mean heart-rate (beats per minute) measures after five 10-s acute interval exercises in the control, endurance-, and strength-training groups. Asterisks (*) represent significant differences (*p* < 0.05).

**Figure 4 biology-11-00961-f004:**
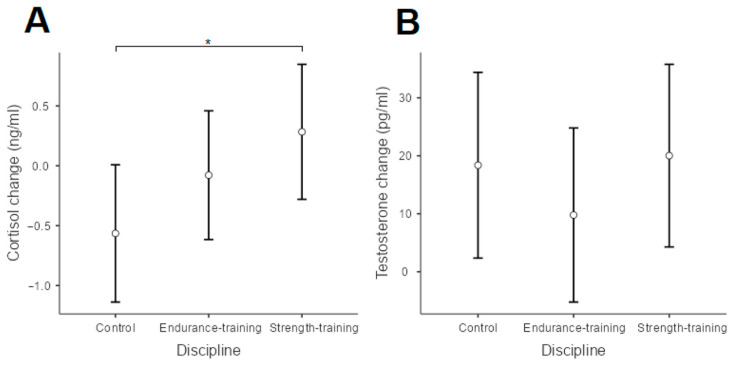
Means and confidence intervals (95%) of the cortisol (**A**) and testosterone (**B**) change after five 10-s acute interval exercises in the control, endurance-, and strength-training groups. An asterisk (*) represents a significant difference (*p* < 0.05).

**Table 1 biology-11-00961-t001:** Participants’ characteristics.

Variables	Endurance Training (*N* = 35)	Strength Training (*N* = 30)	Control (*N* = 31)
Age	20.71 (1.62)	21.53 (1.69)	21.58 (1.96)
Body height (cm)	180.77 (5.55)	183.72 (7.51)	174.45 (5.57)
Body mass (kg)	75.59 (11.34)	80.77 (10.21)	79.81 (9.05)
BMI (kg∙m^−2^)	23.11 (3.16)	24.08 (3.75)	24.89 (3.55)
Physical activity (h per week)	6.60 (3.61)	5.25 (2.81)	4.48 (2.29)
HR_rest_ (beats∙min^–1^)	68.36 (10.80)	68.03 (11.81)	69.41 (9.43)
HR_mean_ (beats∙min^–1^)	152.82 (10.76)	155.83 (13.59)	160.75 (12.12)
Testosterone (pre) Testosterone change (post–pre)	135.46 (62.15) 9.79 (31.07)	129.88 (54.21) 20.02 (47.17)	135.17 (49.05) 18.37 (50.79)
Cortisol (pre) Cortisol change (post–pre)	7.62 (1.75) −0.08 (1.33)	8.07 (1.56) 0.28 (1.73)	8.47 (1.72) −0.57 (1.60)
T/C (pre) T/C change (post–pre)	18.02 (7.46) 49.14 (118.69)	16.32 (6.70) 37.33 (165.49)	16.37 (6.28) −18.58 (126.41)
RPE	5.16 (1.83)	5.60 (2.50)	5.36 (2.19)
Self-reported dominance	3.34 (0.75)	3.31 (0.71)	3.23 (0.63)

Note. Numbers represent means and standard deviations (in brackets). BMI—body mass index (body mass (kg∙m^−2^); HR—heart rate. RPE—ratings of perceived exertion (Borg’s scale), T/C—ratios of testosterone to cortisol.

**Table 2 biology-11-00961-t002:** Summary of the linear regression results with the mean heart rate after five bouts as a dependent variable.

Outcome Variable: Mean Heart Rate	r^2^ = 0.124, F_(5.84)_ = 3.513, *p* = 0.006			
Predictor	β	95% CI	SE	*p*
Discipline				
Endurance–Control	−0.698	[−1.191, −0.204]	3.127	0.006 **
Strength–Control	−0.531	[−1.041, −0.021]	3.229	0.041 *
Testosterone change	0.075	[−0.128, 0.278]	0.030	0.466
Cortisol change	0.233	[0.023, 0.442]	0.883	0.030 *
Self-reported dominance	−0.221	[−0.420, −0.022]	1.817	0.030 *

Note. * *p* < 0.05, ** *p* < 0.01.

**Table 3 biology-11-00961-t003:** Summary of the linear regression results with the cortisol change after five bouts as a dependent variable.

Outcome Variable: Cortisol Change	r^2^ = 0.080, F_(3.86)_ = 2.503, *p* = 0.065			
Predictor	β	95% CI	SE	*p*
Discipline				
Endurance–Control	0.436	[−0.067, 0.939]	0.380	0.089
Strength–Control	0.689	[0.179, 1.199]	0.385	0.009 **
Self-reported dominance	0.028	[−0.178, 0.234]	0.224	0.790

Note. ** *p* < 0.01.

**Table 4 biology-11-00961-t004:** Summary of the linear regression results with the testosterone change after five bouts as a dependent variable.

Outcome Variable: Testosterone Change	r^2^ = 0.156, F_(3.86)_ = 0.711, *p* = 0.548			
Predictor	β	95% CI	SE	*p*
Discipline				
Endurance–Control	−0.174	[−0.692, 0.344]	11.281	0.506
Strength–Control	0.065	[−0.461, 0.59]	11.443	0.807
Self-reported dominance	0.120	[−0.092, 0.332]	6.657	0.264

## Data Availability

The datasets used and/or analyzed during this study are available from the corresponding author on reasonable request.

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
