# Peer review of "Effects of Dominance and Sprint Interval Exercise on Testosterone and Cortisol Levels in Strength-, Endurance-, and Non-Training Men"

_biology, 2022, doi:10.3390/biology11070961_

Round 1
Reviewer 1 Report
This is an interesting study and falls in the Scope of biology of Sports. This reviewer believes that the study is important to scientific literature. However, there's place to improve the manuscript.
Specific commentaries bellow:
Introduction - well written
Methods:
- control group perform 2 times per week physical activity. How much is thee volume and intensity? That may be a limitation of the study, depending on physical activity levels. Is there any chance to influence the results? This should be hilighted in discussion.
- Statistical anaalysis: missing effect sizes; it would be nice to include Bland-Altman plots.
Results
This reviwer believes that Bland Altman plots may improve the analysis and results. Include effect sizes will provide a deeper understanding of the results.
Diacussion
Improve based on new statiscal procedures.
Theorical background was well choosed.
Limitations: was sleep controlled? How it may affect T and C flutuations?
Conclusion:
Well written
Author Response
We would like to sincerely thank the Reviewer for the positive feedback and helpful comments. We focused our efforts strongly on the points made in your letter. As advised, we would like to respond to this opinion based on our careful revision, point by point (see attached file).

Reviewer 2 Report
Dear authors, thanks for your resubmission. Sadly I noticed the same issues. Briefly, in this version Dominance was introduced, with no apparent reason, it was not presented in the Intro, neither any connections, mechanistic/physiological one were introduced. Same stands for the the remaining of items that were not connected as concepts, explain the problem and how this study is driving the field vertically.
Methods are lacking clarity and soundness. The creation of 3 groups is not comparable and the stats are not appropriate to answer the RQ.
Author Response

(The authors gave the same response as above.)

Reviewer 3 Report
Overview
The authors aimed to investigate the effect of a sprint interval exercise on changes in testosterone and cortisol levels and the T/C marker among non-training, endurance-training, and strength-training participants. Findings indicated that the sprint interval exercise protocol performed by endurance-training and strength-training does not cause significant differences in hormonal conditions compared to the control group.
This manuscript is well written, and the topic is original and very interesting.
Below are my specific comments.
Specific comments
Abstract
The abstract summarizes the manuscript correctly and provides the highlights of each section of the document.
Key Words
-Line 45: Replace the keyword "dominance" with another word other than the title. To optimize the search for the manuscript through search engines, it is necessary to enter keywords other than the title.
Introduction
The authors have provided a good summary of the literature in a concise way. The gap in the literature to be filled has been correctly described and the aims and hypotheses formulated are clear.
Materials and Methods
The methodology was clearly explained.
The measurements taken were described correctly.
The statistics used are appropriate.
-One of my concerns to clarify:
Was an a priori power analysis performed to establish the sample size?
Please describe the analysis performed to determine the number of study participants.
Results
The Results section was written correctly.
The tables and figures are explanatory.
Discussion
The authors' discussions and conclusions are justified by the findings made.
Discussion and conclusions are written clearly and precisely. The limitations described are appropriate.
The take-home message is clear.
References
The references cited are relevant and mostly current.
Author Response

(The authors gave the same response as above.)

Reviewer 4 Report
This manuscript is written well, however, this will need to revise just a few points.
1) I did not understand this manuscript's subject of the sample size.
So, could you describe this exactly?
2) Could you show that selection of the subjects using flow chart?
Author Response

(The authors gave the same response as above.)

Round 2
Reviewer 1 Report
Dear reviewers, the manuscript was improved. However, this reviewer did not understand why effect sizes were not included. You justified why not include B-A plots, nothing was mentioned regarding to the effect sizes.
Author Response
Dear Reviewer,
Thank you for pointing out important points in our manuscript. We have revised and, we hope, clarified what was requested. We hope that our responses will meet the reviewer's expectations

Reviewer 2 Report
Dear authors, thanks for submitting again your manuscript.
Sadly the issues were not solved.
First I do not understand the purpose of including dominance in here, especially by using a scale that is not validated. This is something that you did in a previous publication and even though it was published, based on the psychometric principles this is highly inappropriate. First you perform a validation/reliability study to prove the usefulness of the scale and then you use it in studies.
Intro attempts to connect the concepts, much better than the previous times, but still the use of Training and Exercise interchangeably becomes confusing and problematic. SIT is not the same as SIE and since you are measuring hormones this is an issue. Another problem is that you try to compare groups based on their exercise training and this is a complete failure.
The key component is Intensity that influences the T and C levels and not just the mode or duration of the exercise. Therefore the allocation of the participants to the respective groups is misleading. For example you cannot compare a bodybuilder that trains to compete to an event with a person that swims regularly and a person that is your control that does 2 hrs of walking to lose weight and improve cardiometabolic risk factors.
Another thing that is problematic is the Stats. First you report incorrect df for the baseline values to show that groups were similar. Then you use regression with HR to be the Y that does not relate to title and the RQ of the study. Your DVs are T and C and IVs are Dominance and SIE with the Dominance to be mainly Covariate.
Effects of dominance, and sprint interval exercise on testosterone and cortisol levels in strength-, endurance-, and non-training men
This title points to comparisons between 3 groups on the DVs - T and C. The dominance can be the covariate on this. Due to all these I will suggest rejection as findings cannot be supported.
On top of that I found that the in some cases the cited papers were improper used, for example they were discussing about health and dominance and you "interpreted" this as exercise and performance related.

Author Response

(The authors gave the same response as above.)

Reviewer 3 Report
Thank you for your replies!
Author Response
Dear Reviewer,
Thank you for your effor to review our manuscript.